# Non-Contact In-Vehicle Occupant Monitoring System Based on Point Clouds from FMCW Radar

Yixuan Chen [1], Yunlong Luo [1,2], Jianhua Ma [3], Alex Qi [2], Runhe Huang [3], Francesco De Paulis [4] and Yihong Qi [1,2,*]

1   School of Information Science and Technology, Southwest Jiaotong University, Chengdu 610032, China; Yixuan.Chen@my.swjtu.edu.cn (Y.C.); yunlong.luo@my.swjtu.edu.cn (Y.L.)
2   Pontosense Inc., Waterloo, ON N2J 4G8, Canada; alex.qi@pontosense.com
3   Faculty of Computer and Information Sciences, Hosei University, Chiyoda City 102-8160, Japan; jianhua@hosei.ac.jp (J.M.); rhuang@hosei.ac.jp (R.H.)
4   UAq EMC Laboratory, University of L'Aquila, 67100 L'Aquila, Italy; francesco.depaulis@univaq.it
*   Correspondence: yihongqi@gmail.com

**Abstract:** In order to reduce the probability of automobile safety incidents, the in-vehicle occupant monitoring is indispensable. However, occupant monitoring using frequency-modulated continuous wave (FMCW) radar can be challenging due to the interference from passengers' posture, movement, and the presence of multiple people. This paper proposes an improved method for generating point clouds using FMCW radar. The approach involves point cloud clustering, post-processing operations such as segmentation, merging, and filtering of the clustered point cloud to match the actual in-vehicle environment, and a state machine combination step. Experimental results show that the proposed method can achieve high recognition accuracy in scenarios with multiple passengers who are moving and sitting in a relaxed manner.

**Keywords:** FMCW; radar; in-vehicle; occupant monitoring system; presence detection; point cloud; clustering; state machine

## 1. Introduction

It is estimated that over 90% of road traffic accidents are caused by human error. In the United States alone, more than 30,000 people die in fatal crashes each year, a number which increased in 2020 due to the COVID-19 pandemic [1]. Tragically, there have also been incidents of people being trapped in enclosed vehicles, resulting in suffocation and heatstroke. According to survey data released by the organization KidsAndCars in 2021, nearly 1000 children died in such incidents since 1990, with 55% of these accidents caused by parents leaving their children unattended in the car [2]. Seat belts and airbags are essential fixtures inside vehicles that can protect passengers in the event of a collision and that can effectively mitigate injuries. Furthermore, adaptively adjusted seat belts and airbags can provide better protection for passengers of different types and heights [3]. In addition, the presence detection of occupants in a vehicle is now considered an indispensable functionality and will soon become an important criterion for car safety ratings, according to the 2025 European New Car Assessment Program [4]. Among the various safety functions enabled by presence detection, child presence detection (CPD) is crucial for adjusting the fitting of seat belts and airbags and for preventing unattended children from being left inside vehicles. The selection of detection equipment for these functions should be carefully evaluated, weighing the pros and cons of different options and scenarios.

There are two categories of devices that can detect passengers' presence and their sizes: contact and non-contact devices. Contact devices, such as under-seat pressure sensors, are not always effective as they cannot easily distinguish between a human body and a heavy object [5], thus requiring the deployment of many sensors throughout the

vehicle [6]. On the other hand, non-contact solutions, such as passive IR sensors (PIR), can be implemented using various technologies. However, the use of infrared technology based on the pyroelectric principle is limited and cannot distinguish between static human bodies [7]. Moreover, PIR sensors are unable to determine the direction of the signal and thus cannot distinguish between different passengers. Camera-based monitoring systems require appropriate illumination and are also subject to privacy infringements [8]. Sensing devices based on radar, particularly those using frequency modulated continuous wave (FMCW), can obtain target information such as distance, angle, speed, and target size, making them superior when compared to other non-contact sensing technologies. FMCW technology provides access to a large amount of information at a low cost and is well suited for different working conditions, making it highly suitable for in-vehicle detection scenarios.

In-vehicle monitoring is a challenging task due to numerous uncertainties in the application scenario. The limited space and different vibrations from the car, along with the large variability of signal types and levels depending on the type of vehicle, make the monitoring task particularly difficult. The specific function of in-vehicle presence detection is to determine the presence of passengers in each position to support subsequent safety measures, such as reminding passengers to fasten their seat belts or adjusting the airbag ejection parameters. Additionally, the presence detection function aims at preventing children from being left unattended in the car.

One important class of FMCW-based methods for in-vehicle occupant monitoring is based on point clouds. However, there are two main challenges associated with this method. First, it is a challenge to determine the number of passengers corresponding to a large number of point clouds. The posture of passengers may lead to incorrect identification of the number of passengers, e.g., a point cloud generated by a forward-leaning passenger may be similar, in some cases, to the point cloud generated by two passengers. Second, if passengers have large movements inside the vehicle, flickering point clouds can be generated due to multipath effects, which can lead to false recognition by creating false images at locations where no one is present. In this paper, we address these challenges by using clustering and post-clustering processing such as segmentation, merging, and filtering of the clustered point cloud according to the actual in-vehicle environment to reduce the mutual interference of a person's posture and to determine the target position adaptively. Additionally, we adopt a state machine to weaken the interference from a person's motion and to adaptively determine the preexisting state of the target.

Existing point cloud-based vehicle detection systems are prone to false positives or missing targets when multiple passengers make body movements. This paper proposes a new point cloud-based vehicle occupancy detection system that overcomes the problem of low-quality point clouds by utilizing clustering and post-processing techniques, as well as a newly designed state machine method. The innovation lies in the first-time application of clustering and post-processing methods in this scenario. The system was validated in the presence of multiple passengers making body movements. The most significant contribution of this paper is to filter, segment, and merge the point cloud cluster according to the actual distribution of car seats, aiming at the impact of passenger sitting movements and a multi-person environment on point cloud clustering, so as to make the point cloud correctly correspond to the actual target and thus improve the recognition accuracy in the real in-vehicle environment. The proposed clustering and post-clustering processing greatly improve the reliability and stability of the system, whereas the novel presence detection method based on a state machine and point cloud features reduces the interference from passenger movements and adaptively determines the target's preexisting state. Experimental results demonstrate the good performance of the proposed algorithms for occupant detection.

The remainder of this paper is structured as follows: Section 2 outlines recent research progress, and Section 3 introduces the system framework on which this research is based. Section 4 describes the processing flow and principles of the algorithms used in point cloud

clustering and presence detection. The experiments and evaluation results are presented in Section 5, and the paper concludes with a summary and conclusions in the final section.

## 2. Related Works

Currently, various sensing devices are implemented for human target detection; they can be broadly divided into two categories: contact-based and non-contact-based sensing. Contact-based systems typically use pressure sensors and wearable devices, while non-contact devices can be further classified into non-radar and radar-based methods. Table 1 presents the specific features of the aforementioned monitoring solutions and their respective advantages and disadvantages when applied in vehicle scenarios.

**Table 1.** Comparison of different monitoring technologies.

| Category | Technology | Sensor Type | Feature and problems |
|---|---|---|---|
| Contact | Embedded | Pressure sensor [5] | Uses an embedded pressure sensor for estimating the weight on a seat, but it cannot distinguish between heavy objects and human bodies. |
| | Wearable | Motion sensor [6] | Uses machine learning models for human movements recognition, which is inconvenient to operate. |
| Non-contact | Non-radar based | Infrared sensor [7] | Uses a pattern recognition algorithm for detecting the presence of people in indoor environments, but it cannot detect fixed people. |
| | | Camera [8] | Uses DNN for in-vehicle occupant presence detection, but its result is heavily influenced by the light factor. Moreover, privacy is a problem. |
| | | Thermal image sensor [9] | Uses thermal imaging methods to determine the presence of a person and uses image density for estimating the number of persons. Its cost is high. |
| | | Ultrasonic sensor [10] | Used for obtaining vital sign information of static targets; however, it cannot easily handle dynamic targets. |
| | Radar-based | Ultra-wide band radar [11,12] | Uses signal processing techniques for measurement of vital signs such as respiration and heart rate monitoring. It is characterized by high computational complexity and high cost. |
| | | Pulsed coherent radar [13] | Uses vital signs for identifying living organisms in the car. It has experimented with baby simulators. |
| | | Continuous wave radar [14] | Uses micro-Doppler and neural networks for classification of occupancy in vehicles. Its results may be influenced by a person's movement. |
| | | FMCW [15–22] | Uses two types of features, i.e., respiratory signs of a person and output of angle-of-arrival (AoA) algorithms, for classification of occupancy in vehicles. Neural network methods are commonly applied to it. The problem is that it requires enormous computation, thus making it difficult to implement in radar chips. |

Contact devices for human target detection mainly include pressure sensors and wearable devices such as bracelets. For instance, in April 2017, Tao et al. proposed a graphene paper pressure sensor that can be used for respiratory and pulse detection as well as motion detection [5]. However, the main challenge for pressure sensors in a vehicle testing environment is that they are unable to distinguish between heavy objects and human bodies, leading to false judgments. To address this, Márquez et al. combined a smart multisensory bracelet to develop a control platform capable of recognizing human activity and detecting situations that may result in body injury [6].

Non-contact devices can be divided into two categories: radar-based and non-radar based. Among non-radar-based systems, Perra et al. proposed a method that combines thermal imaging from an infrared sensor array with a pattern recognition algorithm to detect the presence of people in indoor environments [7]. The algorithm can count people in a confined indoor environment, similar to that of a car, with high accuracy by detecting

the movement of users with different walking speeds based on different user–sensor distances and the typical temperature ranges of residential environments. However, the algorithm's performance is degraded by different distances, speeds, and thermal noise. Additionally, devices based on infrared systems have inherent limitations due to the pyroelectric effect and cannot recognize stationary targets. Another widely used non-contact system is based on camera images for occupant detection, as presented by Papakis et al. [8]. This system constructs a deep learning neural network to sense body movements and actions and identify the number of people in a vehicle. Alternatively, a thermal imaging camera can also enable the detection of the presence of people indoors, but this method is expensive [9]. Additionally, an ultrasound-based system can detect static human vital signs, but monitoring accuracy decreases in complex in-vehicle environments [10].

Various radar-based systems and methodologies have been developed and are available in the literature. One such system is based on ultra-wideband radar [11], which determines the presence of passengers mainly by analyzing parameters such as signal mainstream and size. This system detects the presence or absence of vital sign signals to detect the presence of people. Signal processing techniques such as the fast Fourier transform (FFT) and variational mode decomposition (VMD) are applied to extract information about vital signs. However, the accuracy of the respiratory cycle estimation is affected by body movements. Lim et al. proposed a method based on pulse radio ultra-wideband radar that feeds the processed signals into a deep neural network [12]. However, the radar used in this method is too large to be embedded in an automotive environment, and it is less robust and challenging to implement in an embedded device or DSP due to the use of integrated learning and multilayer perception techniques. Lazaro et al. proposed a detection method based on peak amplitude difference standard deviation estimation and threshold comparator [13]. The respiration rate is measured from the time interval between two peaks associated with breathing movement. Hyun et al. proposed a Doppler spectrum-based vehicle passenger detection scheme [14]. In their research, they utilized two motion features and one vital sign feature, and then applied machine learning-based identification using a bifurcated decision tree (BDT).

The proposed solution in this work is based on a Frequency Modulated Continuous Wave (FMCW) radar, which is inexpensive, small in size, and easy to deploy. It is equipped with a built-in chip for offline computing and is able to derive data on multiple dimensions of the target, such as distance, angle, speed, and target size due to its signal characteristics. Therefore, a multi-input multi-output FMCW radar is suitable for the in-vehicle scenario. Additionally, Caddemi et al. developed a child recognition system using sawtooth wave radar [15]. In the case of a child in a child seat, the FMCW radar beam is pointed at the seat, and the target distance is calculated by FFT using the distance difference between the presence and absence of the child. Although this scheme is easy to implement, its primary disadvantage is that the radar can only detect one position, and the seat cannot move back and forth during its use unless the threshold for determining the child's presence is reset after the seat movement. In 2021, Song et al. used an FFT-based system to identify the child's presence by analyzing the respiratory band (0.1–0.4 Hz) and the heartbeat band (0.8–1.7 Hz) [16]. Cardillo et al. [17] presents a detection system for a non-contact engine room that utilizes a combination of radar signal processing techniques to detect the presence and location of people inside the vehicle. The experimental results confirm the feasibility and effectiveness of the system, but the study does not test it in a real-world scenario where multiple targets are moving, which is a problem that the paper aims to address. Diewald et al. used Feko software to simulate the directional map of the transceiver antenna and then modeled the full vehicle environment [18]. However, the work concluded that target extraction by phase information is not possible if there are multiple moving targets. Abedi et al. modeled the vehicle and the antenna using a full-wave electromagnetic simulation to find the best installation position of the AWR1443 type of radar [19]. Song et al., on the other hand, used power and Wiener entropy at different distances for the estimation of different seat occupancy situations [20]. Another class of methods relies on the generation

of the target point cloud map using the multidimensional information of FMCW. Zhang et al. proposed a more accurate point cloud generation method by combining the moving target indication (MTI) algorithm and range-Doppler imaging (RDI) using both dynamic and static characteristics [21]. Abedi et al. fed the heat map of AoA's output into different classifiers, including support vector machine (SVM), k-nearest neighbors (KNN), and random forest (RF), to classify passenger occupancy [22]. As mentioned by Abedi [22] in his paper, the group tracking algorithm based on point cloud data may not produce accurate recognition results when passengers are moving. Therefore, he employed machine learning techniques to improve the algorithm's performance. The objective of this paper is to address the challenging scenario of monitoring multiple passengers with movements inside a vehicle. Additionally, the proposed algorithm in this paper is characterized by low complexity and can be easily implemented on embedded systems. In general, there are two main approaches for target detection using FMCW, including passenger presence detection based on their breathing and heartbeat characteristics [14–20] and output of angle of arrival (AoA) [21,22]. However, the former method has limitations due to the impact of the moving vehicle and passengers' movements, which can significantly affect the measurement of vital signs. Similarly, the FMCW-based method of extracting micro doppler features is unsatisfactory in situations where multiple passengers are moving simultaneously, which is the most realistic scenario considered in this paper. Therefore, this paper proposes a point cloud generation method that is less affected by movement and interference. The focus of this paper is not the generation of point clouds, but the post-processing process for low precision point clouds. The purpose of this paper is to propose a detection scheme that can adapt to different sitting positions and movements of passengers based on the low precision point cloud method, to achieve better robustness and improve detection accuracy under the condition of low complexity. The proposed method uses clustering and a state machine to achieve robust presence detection with low computational complexity and ease of deployment on the radar's microcontroller unit (MCU). This method offers better robustness to passengers' actions, and it is characterized by low cost and low latency. Unlike other methods in Table 1, the proposed method has been experimentally demonstrated to be applicable in scenarios that closely resemble a real car environment with moving passengers.

## 3. Layered System Framework for Occupant Monitoring

To describe the proposed solution for occupant monitoring and to facilitate future research on other in-vehicle applications, a layered system framework is proposed which spans from the underlying hardware to the top-level application. The proposed framework, illustrated in Figure 1, comprises four layers: the electromagnetic wave layer, the signal processing layer, the data analytics layer, and the smart application layer.

In the electromagnetic wave layer, the FMCW radar is utilized to gather information on the spatial distribution of people inside the vehicle. By processing the IF signal using Fourier transform-based techniques and combining it with incoming wave direction estimation, various information such as target distance, velocity, angle, and target size can be obtained.

As shown in Figure 2, an FMCW radar transmits and receives a chirp signal, which is later post-processed. The delay between the TX chirp and the RX chirp is

$$\tau = \frac{2d}{c} \tag{1}$$

where $d$ is the distance between the radar and the object, and $c$ is the speed of light. The mixing produces the IF signal, and the frequency is

$$f_c = \frac{\tau}{T_c}B + f_0 \tag{2}$$

where $f_0$ is the starting frequency of the radar; $T_c$ is the TX chirp's frequency rise time; and $B$ is the bandwidth of radar. The IF signal frequency corresponding to the target object

is obtained by converting the time domain IF signal to the frequency domain using the fast Fourier transform (FFT). Thus, the specific distance corresponding to the object is calculated. In addition, the resolution of the FFT-based range estimation is determined by the swept bandwidth **B** of the FMCW system.

$$R_{res} = \frac{c}{2B} \tag{3}$$

| Four-Layers System Framework | | | | Purposes |
|---|---|---|---|---|
| **Smart Application Layer** | | | | Reliable Recognition |
| Presence Detection | Occupant Classification | Driver Monitoring | Person Identification | |
| **Data Analytics Layer** | | | | Machine Learning |
| Clustering | Post-clustering Processing | Feature Extraction | Status Machine | |
| **Signal Processing Layer** | | | | Signal Cleansing |
| ADC Data | Range FFT | Clutter Remove | AoA & CFAR | |
| **Electromagnetic Wave Layer** | | | | Low Noise Hardware |
| RF Transmitter Module | RF Receiver Module | Antenna Array | Antenna Radome | |

**Figure 1.** General layered system framework for the in-vehicle occupant monitoring system.

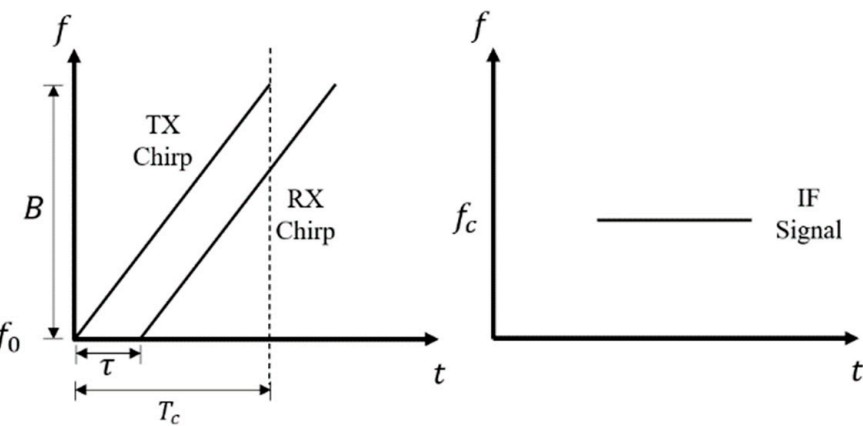

**Figure 2.** The TX chirp, RX chirp, and the IF signal.

In our following experiments, the FMCW signal is configured with 2 GHz swept bandwidth, and the expected range resolution is 7.5 cm, as shown in Equation (3).

Theoretically, this layer can be applied to all kinds of radars. The radar used in this research operates in the frequency band 60–62 GHz with three transmitting antennas and four receiving antennas [23]. Since in-vehicle detection requires angle-of-arrival estimation in two dimensions, the virtual array arrangement is designed as shown in Figure 3. Azimuth and elevation angular resolution are 29 degrees.

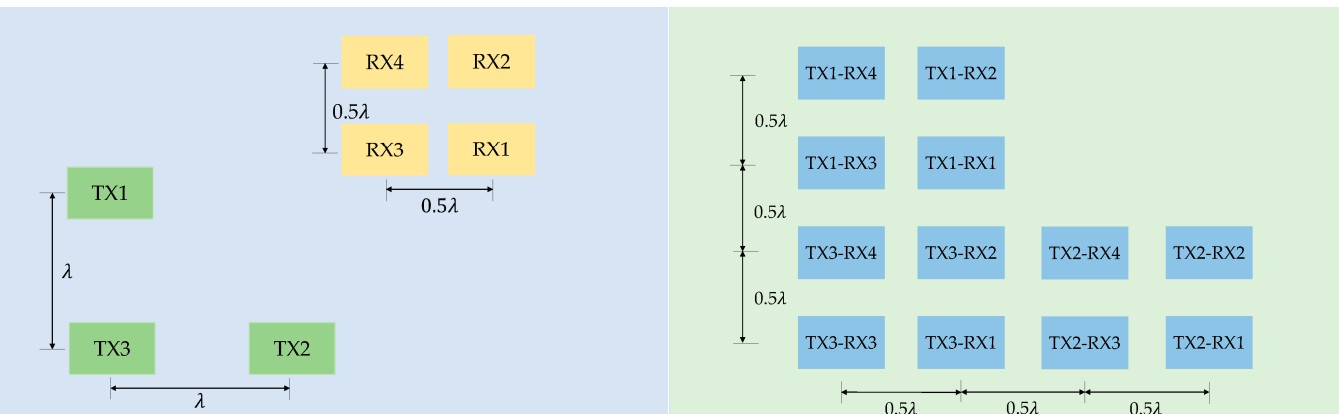

**Figure 3.** Actual antenna array layout (**left**) and virtual antenna array layout (**right**). The virtual array layout describes the virtual channels made by different combinations of transmitting and receiving antennas. Reprinted/adapted with permission from Ref. [23]. 2022, Texas Instruments.

In the signal processing layer, the standard process for generating point clouds is utilized. The IF signals sampled by the ADC are subjected to FFT, followed by clutter removal to eliminate the impact of static reflective surfaces by misusing the mean value in time. Then, the angle of arrival (AoA) is calculated at each frequency using the 2D-CAPON algorithm. In the final step, the dynamic reflective surfaces are extracted using CFAR-CA, and after coordinate conversion, the point cloud is generated [24,25].

The data analysis layer aims at analyzing and extracting multi-dimensional features from the previously generated point cloud data to make decisions about the vehicle occupancy. However, since the point clouds contain noisy data and the positions of seats and people are unpredictable, clustering methods can be used for spatial aggregation. In this paper, various clustering methods are utilized to divide the point clouds into different classes and assign them to specific targets. Although traditional approaches rely solely on the current frame's point cloud to make decisions, they may be inaccurate since the point cloud is in an unstable flickering state due to the influence of moving targets on radar echoes. Large body motions, such as those that occur when passengers change seats or get on and off the car, make the radar detection results unstable because the echoes from moving targets overwhelm those from slightly moving targets, resulting in a poor estimation of the incoming wave direction or large errors in angle estimation. To overcome this limitation, a state machine is employed to describe different states and determine the presence of people in the car through the transfer process between states. This approach enables a more accurate detection result by adapting to the case of moving bodies. Hence, the proposed solution in the data analytics layer adopts a clustering strategy to process the point clouds based on the passenger state machine, as described in the next section. This leads to a self-adaptive judgment step that is beneficial as functional support for the subsequent smart application layer, which improves people counting accuracy.

The top smart application layer mainly offers vehicle occupant presence detection, unattended child prevention, vehicle anti-intrusion, and other possible in-vehicle future applications. The bottom-up framework of the whole system can make full use of the physical radar signals and suitable processing algorithms for adaption to special characteristics of different environments and passengers in various types of vehicles.

## 4. Processing Flow for In-Vehicle Monitoring

The processing flow of the proposed monitoring system consists of three main parts and their main functional modules are shown in Figure 4. The three parts are point cloud generation, clustering algorithm, and presence detection; they are described in the following subsections, respectively.

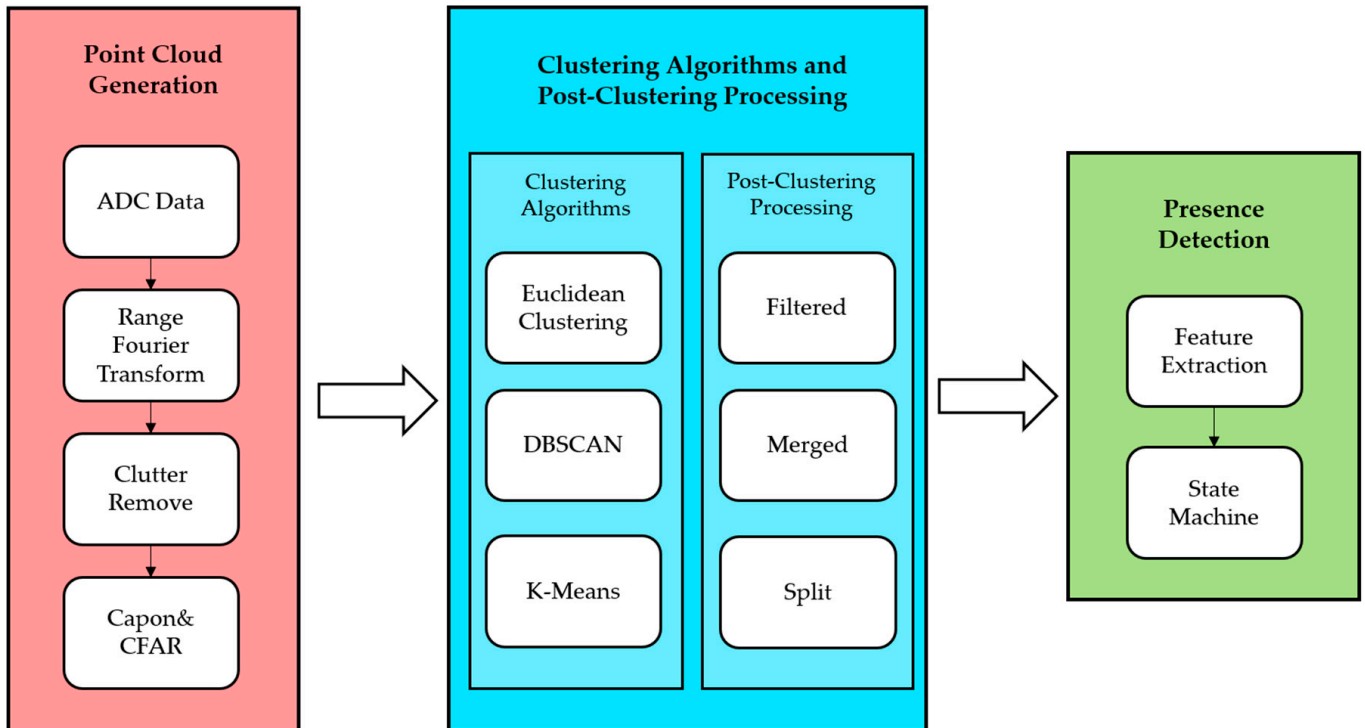

**Figure 4.** Processing flow from point cloud generation to its clustering, and then to presence detection.

### 4.1. Point Cloud Generation and Challenges

The process of obtaining point clouds from the radar is similar to the one outlined in the TI reference processing chain [24]. First, the differential frequency signals are processed by the fast Fourier transform (FFT) to generate the point cloud. As the FMCW radar produces a sawtooth wave, the frequency of the differential frequency signal is proportional to the distance between the reflecting surface and the radar, allowing distance information to be obtained from the result of the Fourier transform. However, static reflecting surfaces such as car seats and chassis may cause interference, and thus, a clutter removal technique is used to eliminate the impact of such surfaces. This technique involves subtracting the result of the FFT for each antenna from its mean value in a period. The incoming wave direction is then estimated using the two-dimensional Capon algorithm [25], which allows for the angular properties of the reflecting surface to be obtained. The CFAR algorithm is used to extract the reflecting surfaces from the Capon results, and the extracted surfaces at different distances are combined to generate point cloud data for the entire space.

The point cloud generation process utilizes the two-dimensional Capon algorithm for angle-of-arrival (AoA) estimation, which is necessary for detecting multiple passengers with different angles at the same distance. However, this also limits the quality of the point cloud. First, the AoA algorithm has an angular resolution of about 29 degrees due to the limited number of antennas, making it difficult for the radar to distinguish between two targets that are too close, as in the scenario where two passengers are sitting next to each other, causing their point clouds to be biased towards one of them. Another problem is the performance of the AoA algorithm when two targets are at the same distance and one is moving more than the other, as the strong reflected signal may drown out the weaker one, resulting in a point loss situation for stationary passengers. Consequently, FMCW-based point clouds have low accuracy and are not stable enough. Therefore, one of the goals of this paper is to mitigate these effects.

The proposed method aims to handle general scenarios that involve passengers in various casual sitting positions and typical movements within a vehicle. The point cloud is first divided based on predefined seating positions, and then presence is determined in each area. However, when a passenger's posture spans multiple areas, the point cloud appears in

multiple areas, making it difficult to determine the number of people in the car. Clustering is used to aggregate point clouds from the same target, thereby avoiding this situation to a certain extent. Furthermore, due to the limitations of the AoA estimation algorithm, the motion of targets can overpower weaker targets, resulting in errors in determining the presence of people. To increase stability and mitigate the impact of body movement, this paper utilizes a state machine approach.

### *4.2. Point Cloud Clustering Principle*

The point cloud data obtained from an FMCW radar in TI's reference design [24] is used directly for occupant detection. However, our experiments have shown that reliable detection results are challenging to achieve, especially when more than two people coexist in a car. This is because the point cloud obtained may not reflect the actual target with high confidence. The points generated by the multipath effect of the target, the points misjudged by the target movement, and the points generated by other objects in the vehicle that do not remain stationary can be confused with the points of the target itself. Therefore, it is necessary to use the distribution characteristics of the point cloud for possible interference removal. Additionally, the situation in which a passenger is sitting in the middle of the area dedicated to a specific seat may be difficult to handle. In such cases, the spatial aggregation property of the point cloud needs to be exploited using the clustering method. We propose the application of clustering on the obtained point cloud to reduce the misjudged and confusing points. Figure 5 shows an example including the original point cloud, the one after clustering, and the one by post-processing of the clustered points obtained by the proposed implementation. Where the positive direction of the x-axis faces to the left of the forward direction of the car, the positive direction of the y-axis faces to the rear of the car, and the positive direction of the z-axis faces to the bottom of the car. It can be seen from the figure that clustered and post-processed point clouds can better distinguish targets, paving the way for subsequent passenger presence determination. In this research, we use three clustering algorithms, and their basic principles are briefly described below.

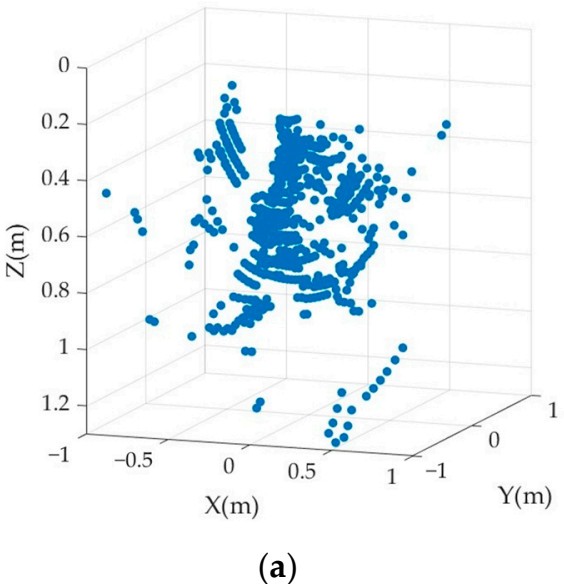
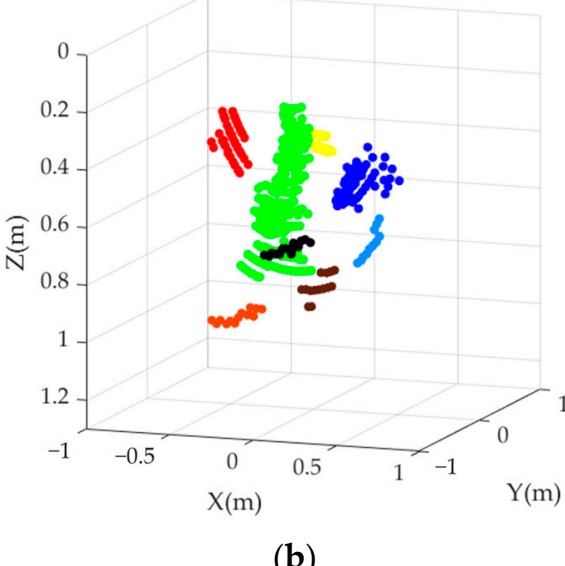

**(a)**  **(b)**

**Figure 5.** *Cont.*

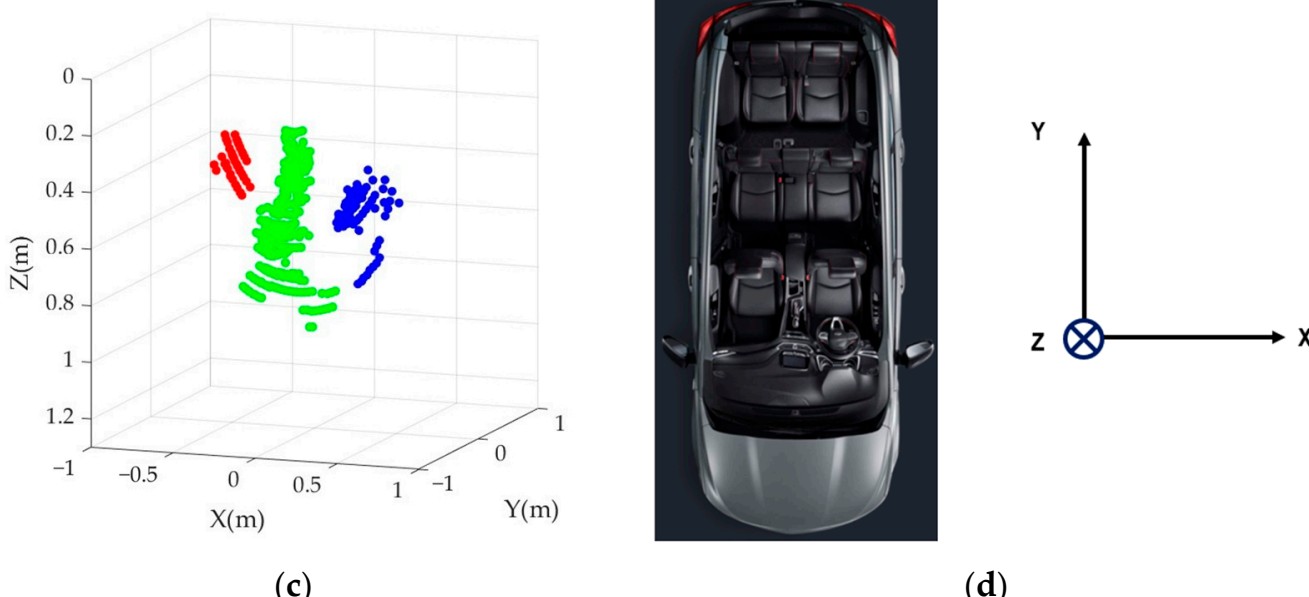

<div align="center">(<b>c</b>)</div> <div align="center">(<b>d</b>)</div>

**Figure 5.** Point clouds for a use case with three passengers in the same row: (**a**) origin point cloud; (**b**) the point cloud after clustering; (**c**) the point cloud after post-processing; (**d**) the positive direction of each coordinate axis. Different colors represent point clouds grouped into different categories.

### 4.2.1. Euclidean Clustering Principle

The Euclidean clustering algorithm employs Euclidean distances to measure the degree of target aggregation and divides all points into non-divisible sets [26]. During this process, the point cloud distribution is used to construct a KDTree in a 3D space, which is equivalent to dividing the 3D space into distinct parts. The KDTree storage structure reduces complexity compared to linear lookup with direct traversal.

### 4.2.2. DBSCAN Principle

DBSCAN is a density-based clustering algorithm that assumes that categories can be determined by the proximity of sample distribution [27]. Samples of the same category are expected to be closely related to each other, meaning that any sample of a given category should not be too far away from the other samples in the same category. By classifying all closely related samples into different categories, the final result of all clustering categories is obtained. This type of density clustering algorithm is widely used in various fields, such as image segmentation and object tracking.

### 4.2.3. K-Means Principle

The k-means algorithm is a distance-based clustering algorithm that uses distance as an evaluation indicator of similarity. In other words, the closer two objects are, the more similar they are considered to be. The k-means algorithm aims to find k clusters in the given data set. It gets its name from the fact that it finds k different clusters, and the center of each cluster is calculated using the mean of the values contained in the cluster. The number of clusters k is specified by the user, and each cluster is described by its centroid, which is the center of all points in the cluster [28].

### 4.2.4. Post-Clustering Processing of Clustered Results

The primary contribution of this paper is in this section, which proposes the use of post-processing methods to handle misclustering and merging of point clouds, allowing for better alignment of the point cloud with real passenger targets and ultimately resulting in improved recognition accuracy. The clustering process may produce results that fall into one of the following categories, as illustrated in Figure 6 based on our experiments: (1) The

point cloud generated by the target's multipath effect and seat vibration has a specific spatial distribution; (2) A target's points may be separated into multiple clusters, such as the head and legs forming two clusters due to the seat blocking the body; (3) The points of two targets may be partially merged when two people are sitting too close together, making it look as a single cluster, especially when their shoulders are almost touching.

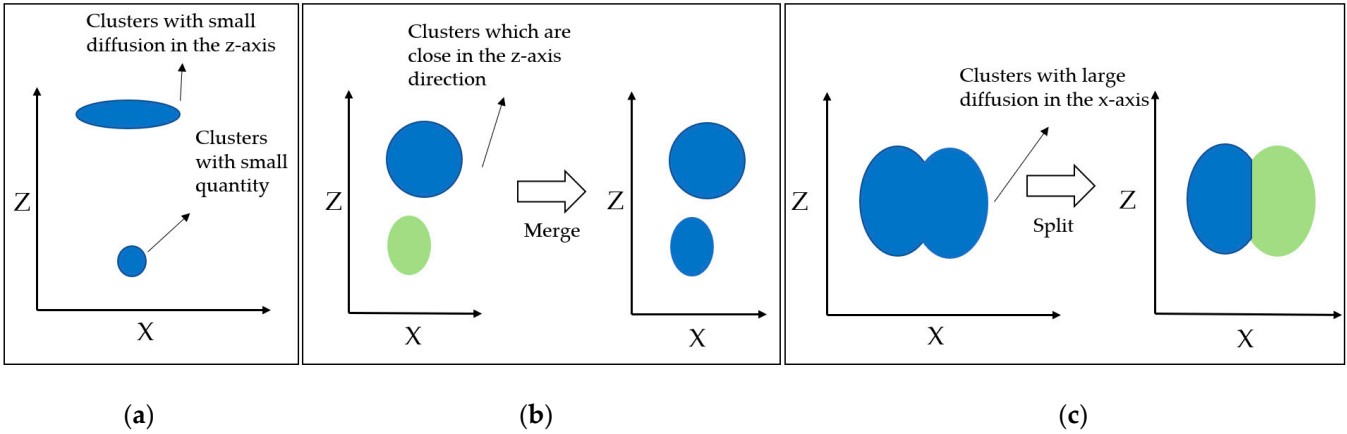

**Figure 6.** Target clusters for post-clustering processing rules: (**a**) clusters with small diffusion in the z-axis and clusters with small quantities will be filtered; (**b**) clusters that are close in the z-axis direction will be merged; (**c**) clusters with large diffusion in the x-axis will be split. Different colors represent point clouds grouped into different categories.

To improve the clustering results, post-clustering processing is proposed to check and handle the three situations described above. The post-processing including filtering, merging, and splitting is applied according to the following three rules.

Rule 1 aims at filtering out point clouds that do not belong to the passengers but are caused by the car's internal environment. Two types of objects are targeted for filtering: the plane at the top of the seat backrest and the car's chassis, which produce thin point clouds on the z-axis due to engine vibrations. Physically, the thickness of these reflective surfaces is minimal, resulting in a variance of less than 14 cm on the z-axis for 97.5% of the point clouds they generate, according to experimental data. The other type of filtering object is small clusters of point clouds generated by the multipath effect in the car. Since their aggregation characteristics are inferior and their number is small, they are not considered to be passengers' point clouds. Point clouds containing less than 20 points are filtered out by the system.

For rule 2, the primary purpose is to handle situations where the passenger's torso is obscured, leading to the passenger being clustered into two separate clusters. Figure 7 shows a top view of three consecutive seats in the second row of the car. The blue color represents the reflection point cloud of the passenger's head, and the green color represents the reflection point cloud of the passenger's legs. When the two clusters are located in the same area and the distance between the two cluster centers is less than the threshold $T_{dis}$, they are merged into a single cluster. $T_{dis}$ is calculated using equation 4, and in the actual experiment, $T_{dis}$ is set to 0.877 m.

$$T_{dis} = \frac{1}{2}\sqrt{\left(\frac{1}{2}Len_{seat}\right)^2 + \left(\frac{1}{2}Wid_{seat}\right)^2} \tag{4}$$

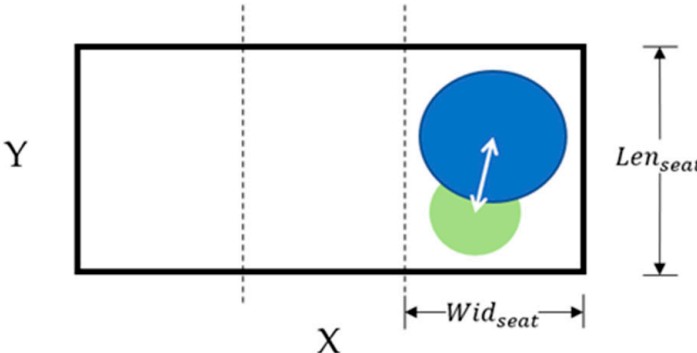

**Figure 7.** The top view of the Figure 6b case. The three rectangles divided by the dotted line are the top view of the three seats. The blue point cloud is the part of the passenger's torso, and the green point cloud indicates the reflection of the passenger's legs. The white arrows indicate the distance between the centers of the point clouds. Different colors represent point clouds grouped into different categories.

For rule 3, the goal is to handle situations where multiple individuals are clustered into one class. Figure 8 depicts a top view of a three-seat configuration in the second row of the car, where two passengers are combined into one class. If $Dist_X > 1.5 Wid_{seat}$, we regard them as two targets and divide them at the x coordinate where the point density is the lowest. In the actual experiment, this threshold was set to 0.778 m.

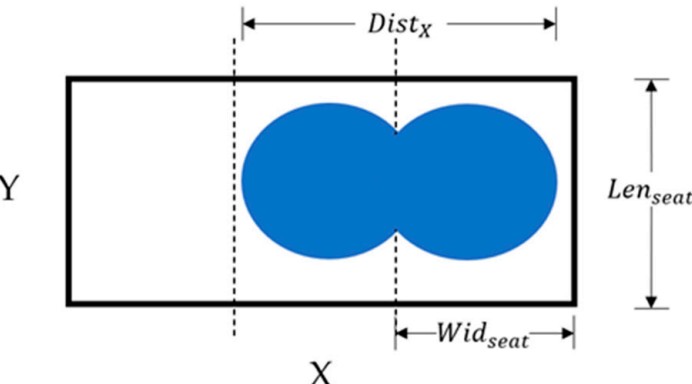

**Figure 8.** The top view of the Figure 6c case. The three rectangles divided by the dotted line are the top view of the three seats. The blue point cloud indicates two passengers that are clustered into one class.

The proposed method improves the clustering process based on the characteristics of point cloud data generated by vehicle occupants. Similar to how the unsupervised ISO-DATA algorithm improves k-means clustering, the clustering results are further segmented and merged to enhance the match between the final clustering results and the actual targets in the vehicle.

### 4.3. Presence Detection Based on a State Machine Diagram

#### 4.3.1. Feature Extraction

The presence detection within a vehicle requires the determination of the presence or absence of passengers in each seat. After the clustering process is completed, the point cloud at each location may not be stable due to the radar echoes of the moving target. For example, a large motion of a target will hide slowly moving targets, the strong echoes will mask the weak echoes. Therefore, the radar detection results are certainly unstable when passengers are making seat changes, or during the process of getting on and off the vehicle. This is mainly due to the poor estimation of the incoming direction for the echoes from

moving targets, since targets at the same distance but with different motion amplitudes will result in the failure of the extraction of the reflected surface points or large errors in angle estimation. Therefore, it is inaccurate to rely on the point cloud data of a single frame only for the determination of the results. As shown in the Figure 9, a case example is considered where a single person is getting on and off the car, and the number of points of the class corresponding to this seat shows a distinctive feature. The different states correspond to different point characteristics. Therefore, the state machine can be considered appropriate to model the passenger's state.

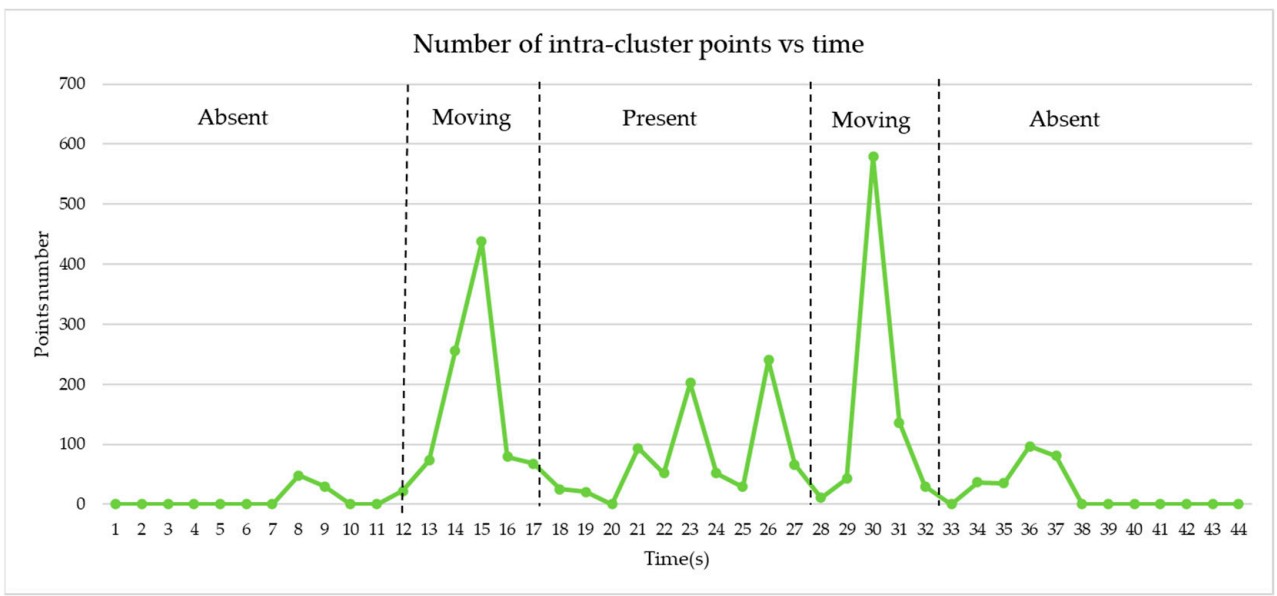

**Figure 9.** Case of a single person getting on and then getting off the car. The figure presents the number of points within the cluster corresponding to the occupied seat as a function of the test person's actions.

### 4.3.2. State Machine

Based on the previous discussion, the states of each location can be categorized as present, absent, or moving. Figure 10 illustrates these three states and their potential state transitions, which are similar to TI's presence detection zone state machine [24]. However, our system's state transfer conditions are based on point sets after clustering and post-processing, which are substantially different from TI's state transfer conditions. The pseudo-code below illustrates the algorithm of the presence detection based on the state machine. Avg_cnt[i] denotes the average number of points over time, cnt[i] denotes the number of points in class i, and T1, T2, and T3 represent the thresholds. T1, T2, and T3 are primarily determined by analyzing and fitting experimental data, as shown in Figure 11. T1 is determined by the cnt value that corresponds to the intersection of the normal fit curves for the ABSENT and PRESENCE states, and T3 is determined by the cnt value that corresponds to the intersection of the curves for the PRESENCE and MOVING states. This approach achieves a statistically significant balance in classifying the states. T2 is the right boundary from the 99% confidence interval of the ABSENT state curve. In the experimental setup validated in this paper, T1, T2, and T3 are set to 32, 52, and 292, respectively. The specific algorithm is described in Algorithm 1.

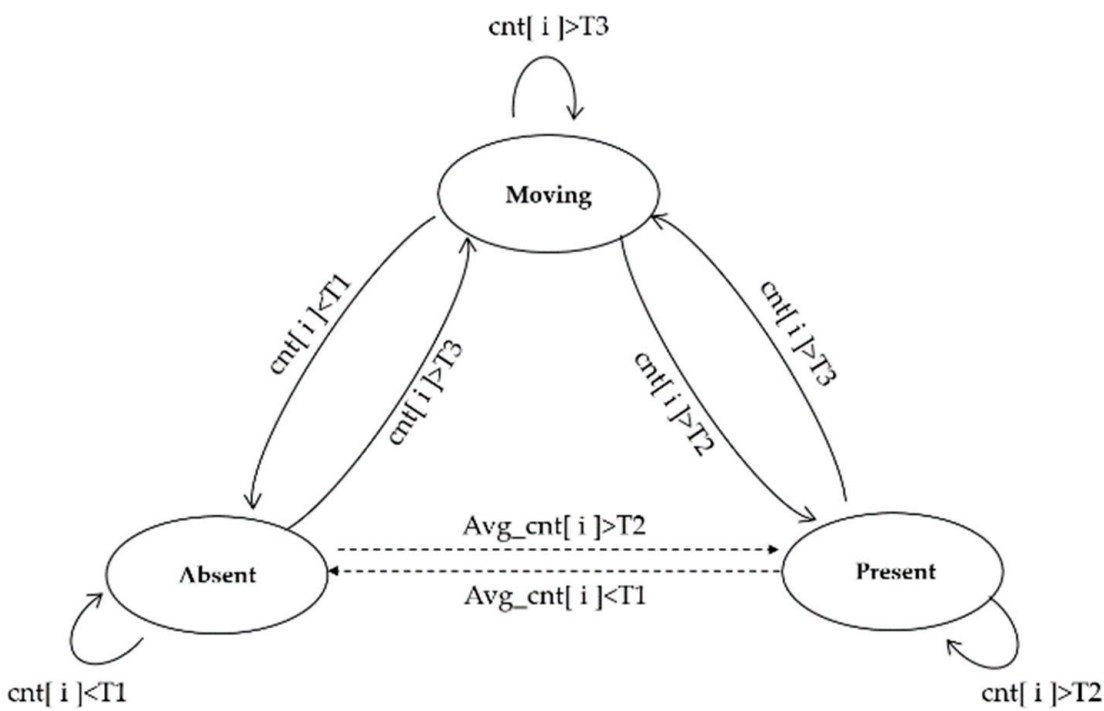

**Figure 10.** Presence state transfer diagram of a passenger.

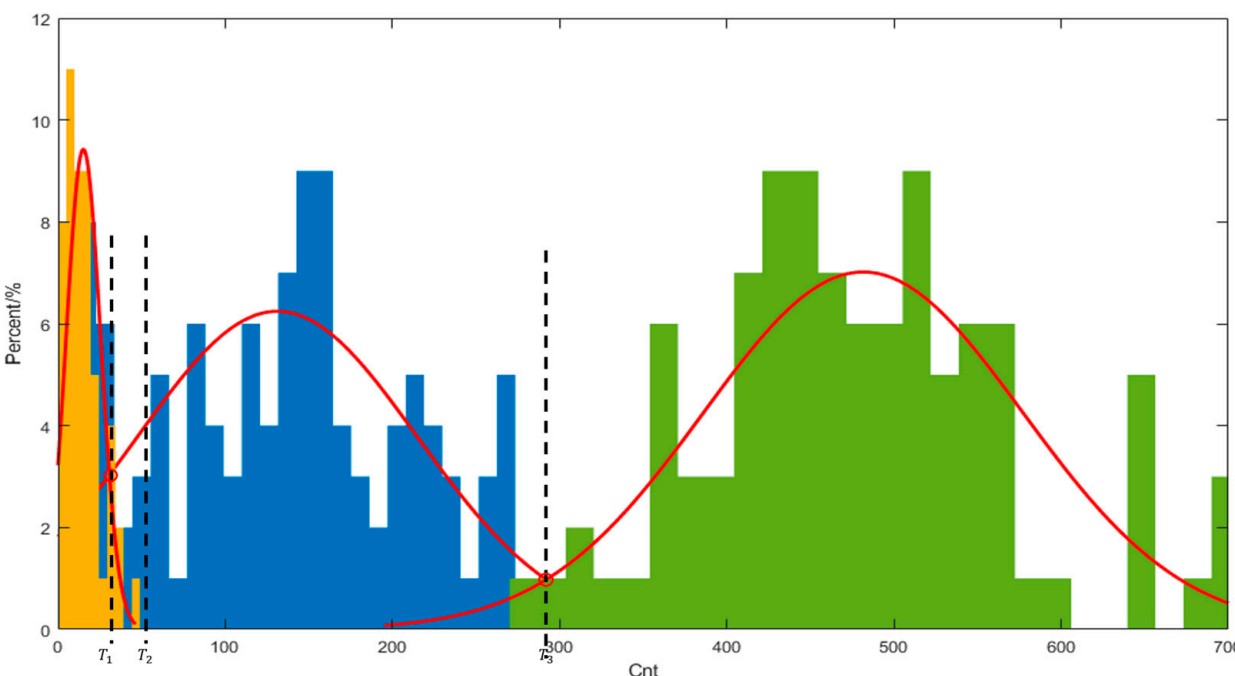

**Figure 11.** The histograms of the respective frequency distributions of cnt in the ABSENT, PRESENCE, and MOVING states and their fitted curves are shown in the figure.

The solid line in Figure 10 represents the primary transfer path, whereas the dashed line represents the secondary transfer path. In real-world scenarios people move in and out of their seats, leading to the transition between present and absent states at each location, thus meaning that the transition between present and absent states must go through the movement state on the primary path. Additionally, the clustered result can be used as an event to trigger the target motion. During the state transfer, the state of the previous moment and the distribution of point clouds at the current moment jointly determine the state of the current moment.

| **Algorithm 1** Presence detection based on state machine |
|---|

```
Initialize model parameters
if Avg_cnt[i]>T2 then
current_state = Present
else
current_state = Absent
while True
do
if current_state = Present then
if cnt[i]>T2 then next_state = Present
    elseif cnt[i]>T3 then next_state = Moving
    elseif Avg_cnt[i]<T1 then next_state = Absent
elseif current_state = Absent then
if cnt[i]<T1 then next_state = Absent
    elseif cnt[i]>T3 then next_state = Moving
    elseif Avg_cnt[i]>T2 then next_state = Present
elseif current_state = Moving then
if cnt[i]>T3 then next_state = Moving
    elseif cnt[i]>T2 then next_state = Present
    elseif cnt[i]<T1 then next_state = Absent
current_state = next_state
endwhile
```

## 5. Experiment and Evaluation

This section presents a series of experiments conducted to demonstrate the effectiveness of the proposed processing flow and associated algorithms. The following subsections describe the experimental environments and several test cases used to apply the proposed method. The resulting evaluation results of the occupant presence detection are presented and discussed.

### 5.1. Experimental Environment and Test Cases for Data Acquisition

The experiments were carried out using TI's IWR6843AOP [23] for data acquisition on a Honda CRV. The system has the following features: a range resolution of 7.5 cm, azimuth and elevation angular resolution of 29 degrees, and a frame rate of 5 frames per second. The subsequent data processing and occupant detection were performed on a PC with an Intel i7-9750H processor operating at 2.60 GHz, 24 GB of RAM, and Windows 11 Professional. The algorithms were implemented using both Python and MATLAB. MATLAB was used for the initial point cloud data analysis, and the subsequent implementation of the whole system was based on python. All the parts are handled by the source code, except for the DBSCAN part in python, which uses functions from the sklearn library. The radar outputted the FFT results via serial communication, which were then stored and reused on the PC. The occupant information considered in the experiments is presented in Table 2.

**Table 2.** Personal information of experiment participants.

| Participant | Height (cm) | Weight (kg) | Age (Years Old) | Gender |
|---|---|---|---|---|
| Adult A | 178 | 60 | 24 | Male |
| Adult B | 175 | 55 | 23 | Male |
| Adult C | 165 | 50 | 23 | Female |
| Child A | 139 | 31 | 9 | Female |
| Child B | 118 | 26 | 6 | Male |

The experimental setup, as illustrated in Figure 12, was used to verify the algorithms proposed in this paper through three types of experiments. These experiments were performed based on two different test cases: static adults and dynamic adults. In the static case, presence detection was performed for zero to five passengers, considering all possible combinations of four different sitting positions and various passenger distributions. On the

other hand, in the dynamic case, the presence of passengers was detected as they boarded and disembarked the vehicle and changed seats, with varying numbers of passengers. Each test case lasted for 1 min and was repeated 3 times, with data output occurring every 250 ms. The results were presented as presence detection outcomes for each seat, and the accuracy was calculated as the percentage of correct determinations out of 240 evaluations.

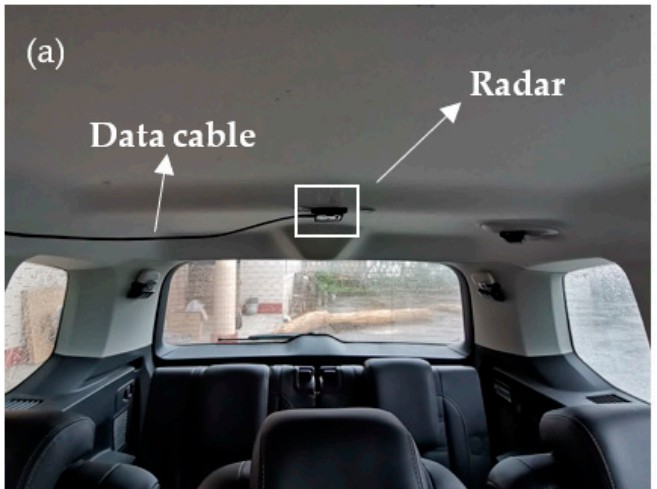
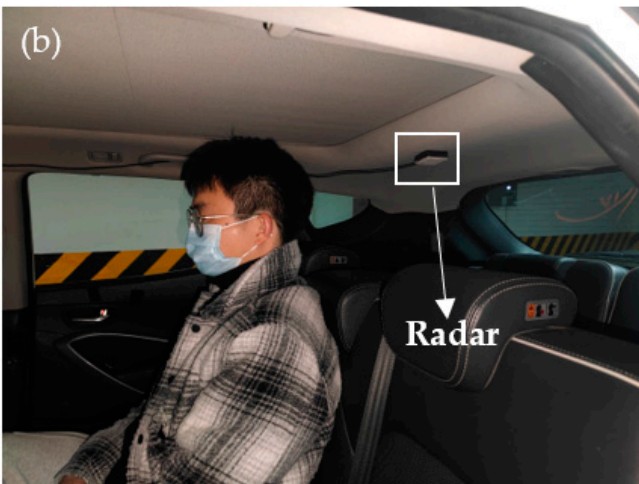
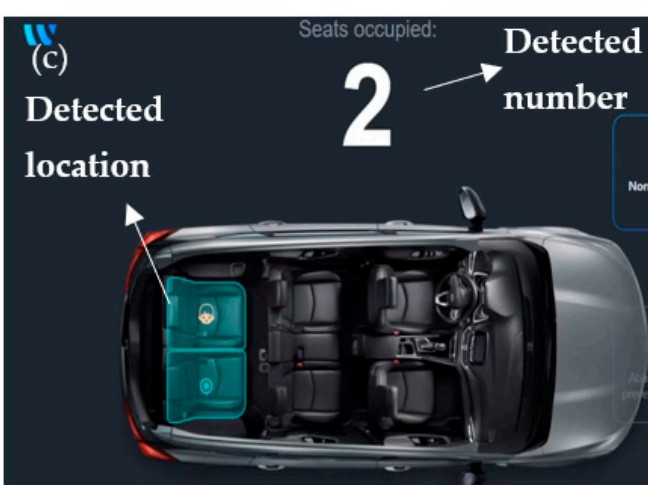
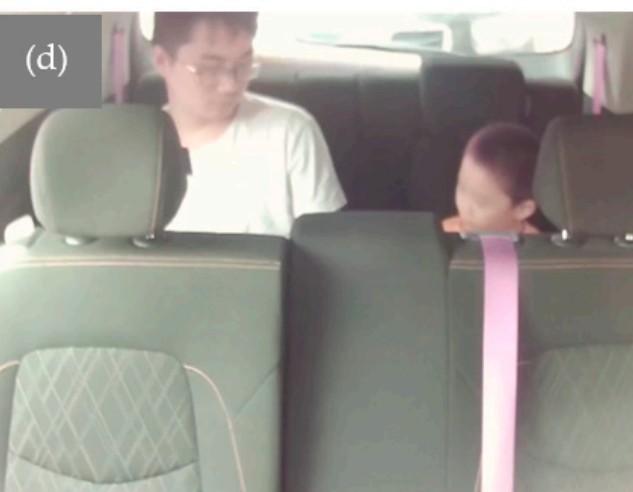

**Figure 12.** Radar placement and experimental environment: (**a**) front view of the radar mounting position; (**b**) left view of passenger and radar; (**c**) the interface where the results are presented. Where the number of people is marked in the upper part and the location of the passengers are marked in the vehicle (**d**) live in-vehicle view of the test case with one adult and one child in the third row.

### 5.2. Evaluation Results

The following sections present the results of the three types of test cases, with accuracy calculated by dividing the number of correct determinations by the total number of evaluations.

### 5.2.1. Presence Detection Results of Static Test Cases

For the static case, we detect the presence of one to five passengers in different combinations of four sitting positions and position distributions. We use the five testers listed in Table 2, with each test lasting 1 min and repeated three times. Accuracy is calculated by dividing the number of frames with the correct number of decisions by the total number of decisions. Table 3 shows the detection accuracy results for the static test cases.

**Table 3.** Presentation detection results of different clustering methods in static use cases.

| | Use Cases | Euclidean | | | | | Detection Accuracy (%) DBSCAN | | | | | K-Means | | | | |
|---|---|---|---|---|---|---|---|---|---|---|---|---|---|---|---|---|
| Test ID | Number of People—Sitting Posture | 1 | 2 | 3 | 4 | 5 | 1 | 2 | 3 | 4 | 5 | 1 | 2 | 3 | 4 | 5 |
| 1 | 1-normal | 100.0 | | | | | 100.0 | | | | | 100.0 | | | | |
| 2 | 1-leaning forward | 100.0 | | | | | 100.0 | | | | | 100.0 | | | | |
| 3 | 1-leaning left | 100.0 | | | | | 100.0 | | | | | 97.6 | 2.4 | | | |
| 4 | 1-leaning right | 100.0 | | | | | 100.0 | | | | | 100.0 | | | | |
| 5 | 2-normal | | 100.0 | | | | 8.0 | 92.0 | | | | 13.1 | 85.1 | 1.8 | | |
| 6 | 2-leaning forward | | 97.6 | 2.4 | | | 4.4 | 95.6 | | | | 1.3 | 85.7 | 13.0 | | |
| 7 | 2-leaning left | 0.6 | 97.6 | 1.8 | | | 7.8 | 91.6 | 0.6 | | | 0.6 | 85.6 | 13.8 | | |
| 8 | 2-leaning right | | 97.1 | 2.9 | | | 8.6 | 91.4 | | | | 14.4 | 85.6 | | | |
| 9 | 3-normal | | 3.6 | 96.4 | | | | 11.3 | 88.7 | | | | 19.6 | 80.4 | | |
| 10 | 3-leaning forward | | 2.6 | 95.2 | 2.2 | | | 14.3 | 85.7 | | | | 21.4 | 77.4 | 1.2 | |
| 11 | 3-leaning left | | 3.6 | 92.9 | 3.6 | | | 15.5 | 84.5 | | | | 17.3 | 79.2 | 3.6 | |
| 12 | 3-leaning right | | 4.6 | 92.3 | 3.1 | | | 13.0 | 85.2 | 1.8 | | | 16.6 | 79.3 | 4.1 | |
| 13 | 4-normal | | | 5.4 | 92.2 | 2.4 | | | 16.3 | 83.7 | | | | 23.5 | 74.1 | 2.4 |
| 14 | 4-leaning forward | | | 6.8 | 90.2 | 3.0 | | 3.0 | 12.7 | 84.3 | | | 0.7 | 27.0 | 69.3 | 3.0 |
| 15 | 4-leaning left | | 1.2 | 5.6 | 90.7 | 2.5 | | 4.6 | 10.8 | 84.6 | | | 0.8 | 24.5 | 72.2 | 2.5 |
| 16 | 4-leaning right | | | 5.9 | 89.7 | 4.4 | | 7.8 | 11.9 | 80.4 | | | | 1.9 | 71.0 | 27.1 |
| 17 | 5-normal | | | | 10.7 | 89.3 | | | 5.0 | 21.3 | 73.7 | | | 15.4 | 16.7 | 67.9 |
| 18 | 5-leaning forward | | | 3.2 | 9.6 | 87.2 | | | 9.2 | 16.4 | 74.4 | | 5.3 | 13.4 | 14.6 | 66.7 |
| 19 | 5-leaning left | | | 5.1 | 9.6 | 85.3 | | | 8.6 | 19.0 | 72.4 | | 0.7 | 13.7 | 19.6 | 66.0 |
| 20 | 5-leaning right | | | 7.5 | 8.2 | 84.2 | | | 5.8 | 16.8 | 77.4 | | | 14.7 | 18.2 | 67.1 |

It is worth noting that none of the test cases were misjudged as having no passengers, which is of great importance in preventing situations where children are forgotten in the car. The clustering methods proved adaptable to different seating arrangements and produced high accuracy rates. The Euclidean clustering and post-clustering processing method proposed in this paper achieved an accuracy of over 90% in various situations with up to four people. However, when there were five passengers, the main obstacle to accuracy was the point cloud quality in addition to passenger posture. In the case of multiple targets, the point cloud often missed the target due to antenna limitations and angle-of-incidence algorithm resolution, resulting in greater accuracy errors with fewer passengers than with more. Comparing the results of different clustering methods, the DBSCAN method tended to group connected point clouds into a single category, leading to target loss for passengers who were close together due to incorrect categorization. Therefore, the determination of lost passengers was often incorrect in cases with multiple passengers. Meanwhile, the k-means method was not stable enough in choosing k values, leading to classification errors and poor clustering robustness. Figure 13 depicts a top view of a point cloud with three individuals leaning to the right in three back seats, with two people being in close proximity. The left figure shows the original point could characterized by unclear clustering, thus not allowing to accurately identify the number of people. However, the proposed post-processing steps, once applied, are able to ensure the correct classification of the point cloud into the appropriate class, thus allowing the accurate identification of the three individuals in the car. Figure 14 demonstrates the effectiveness of the proposed processing methodology even when the car is fully loaded. The five people in the car can be clearly identified only after the post-processing step, with the developed system that can accurately categorize point clouds moving from the raw results in Figure 14a to the clustered point clouds in Figure 14b. The results of different clustering methods in Table 3 are summarized in Figure 15 for a better and more intuitive overview. Based on these results, the Euclidean clustering method proposed in this paper is the most suitable point cloud clustering method for in-vehicle environments.

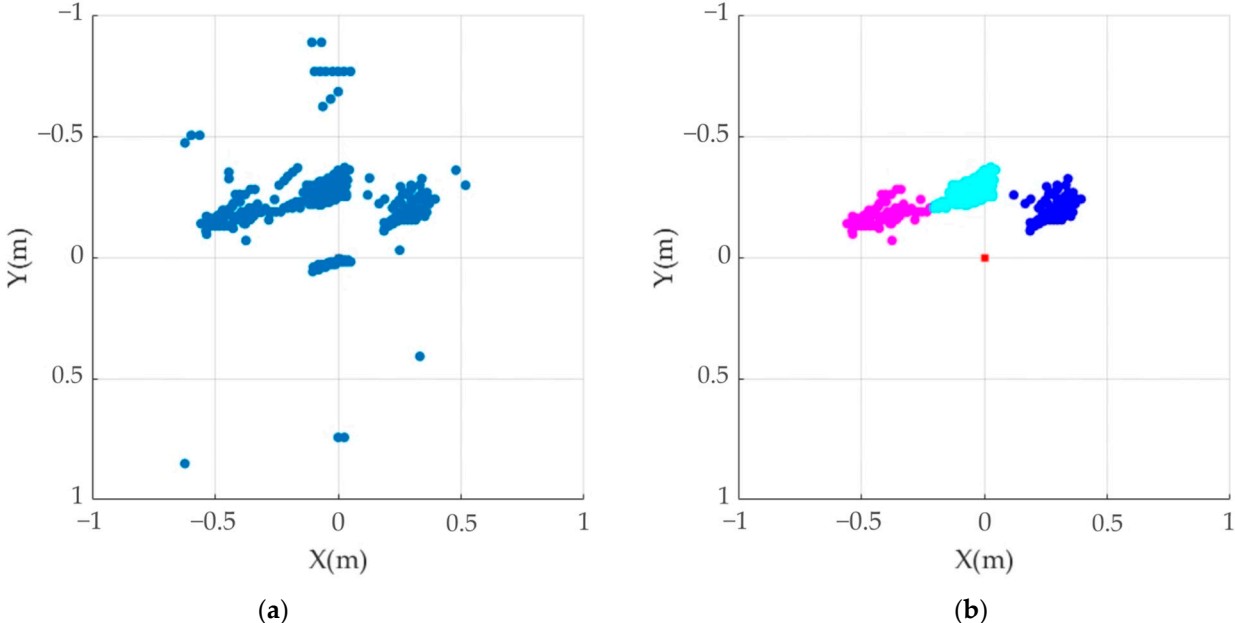

(**a**)                                                       (**b**)

**Figure 13.** This image depicts the top view of a point cloud in a static use case involving three people leaning right in the back row. The original point cloud image is on the left (**a**), whereas the right figure (**b**) shows the result of the clustering and post-processing steps. Different colors are used to represent different categories of point clouds, with the positive *Y*-axis pointing towards the front.

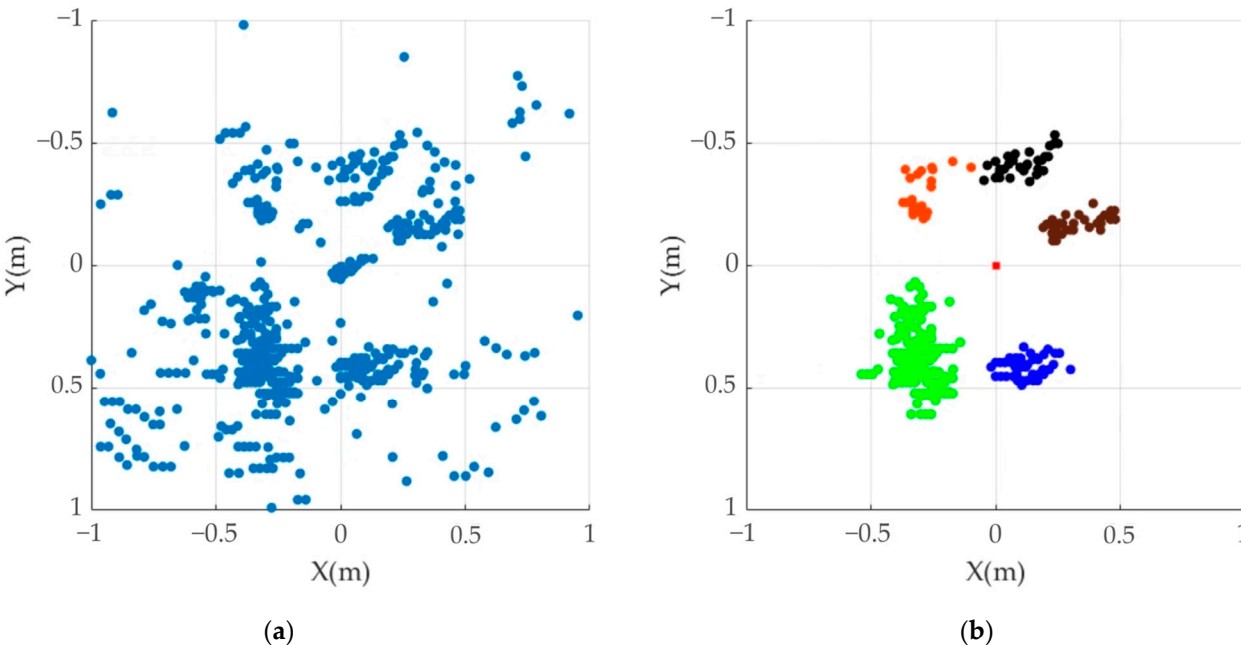

(**a**)　　　　　　　　　　　　　　　　　　(**b**)

**Figure 14.** This image depicts the top view of a point cloud in a static use case, tilted to the right by five people leaning forward in the car. The original point cloud image is on the left (**a**), while the right (**b**) shows the result of clustering and post-processing. Different colors are used to represent different categories of point clouds, with the positive *Y*-axis pointing towards the front.

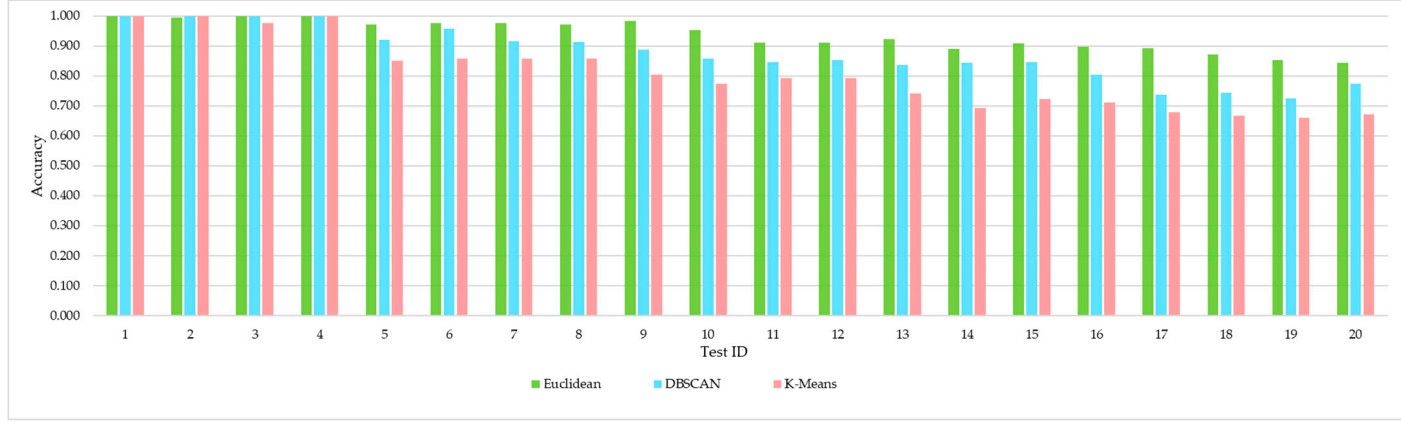

**Figure 15.** Accuracy of different clustering algorithms applied in presence detection function in different use cases.

In this paper, the effectiveness of the clustering post-processing step is verified to improve the detection accuracy of multiple passengers. The results presented in Table 4 compare the accuracy obtained with and without the post-processing step. It can be observed that the average accuracy for four passengers increased from 90.7% to 92.7%, and for five passengers, it increased from 86.5% to 90.4%. Therefore, the effectiveness of the post-processing step can be illustrated. The significant improvement is mainly attributed to the problems described in Section 4, which are likely to occur in multi-passenger scenarios where it is impossible to require passengers to sit upright at all times. The random sitting posture can result in mis-segmentation and adhesion of point cloud clusters, and the clustering post-processing step is effective in addressing this issue.

Table 4. Presentation detection results with or without clustering post-processing in static use cases.

| Use Cases | | Detection Accuracy (%) | | | | | | | | | |
| | | No Cluster Post-Processing | | | | | Cluster Post-Processing | | | | |
| Test ID | Number of People—Sitting Posture | 1 | 2 | 3 | 4 | 5 | 1 | 2 | 3 | 4 | 5 |
|---|---|---|---|---|---|---|---|---|---|---|---|
| 1 | 1-normal | 100.0 | | | | | 100.0 | | | | |
| 2 | 1-leaning forward | 100.0 | | | | | 100.0 | | | | |
| 3 | 1-leaning left | 100.0 | | | | | 100.0 | | | | |
| 4 | 1-leaning right | 100.0 | | | | | 100.0 | | | | |
| 5 | 2-normal | | 100.0 | | | | | 100.0 | | | |
| 6 | 2-leaning forward | | 97.6 | 2.4 | | | | 98.5 | 1.5 | | |
| 7 | 2-leaning left | 0.6 | 97.6 | 1.8 | | | 0.3 | 98.3 | 1.4 | | |
| 8 | 2-leaning right | | 97.1 | 2.9 | | | 0.2 | 97.8 | 2.0 | | |
| 9 | 3-normal | | 3.6 | 96.4 | | | | 1.8 | 97.6 | 0.6 | |
| 10 | 3-leaning forward | | 2.6 | 95.2 | 2.2 | | | 1.4 | 96.0 | 2.6 | |
| 11 | 3-leaning left | | 3.6 | 92.9 | 3.6 | | | 2.7 | 94.2 | 3.1 | |
| 12 | 3-leaning right | | 4.6 | 92.3 | 3.1 | | | 3.3 | 93.5 | 3.2 | |
| 13 | 4-normal | | | 5.4 | 92.2 | 2.4 | | | 3.2 | 92.6 | 4.2 |
| 14 | 4-leaning forward | | | 6.8 | 90.2 | 3.0 | | | 5.5 | 92.7 | 1.8 |
| 15 | 4-leaning left | | 1.2 | 5.6 | 90.7 | 2.5 | | | 4.5 | 93.1 | 2.4 |
| 16 | 4-leaning right | | | 5.9 | 89.7 | 4.4 | | | 2.6 | 92.4 | 5.0 |
| 17 | 5-normal | | | | 10.7 | 89.3 | | | 4.6 | 4.8 | 90.6 |
| 18 | 5-leaning forward | | | 3.2 | 9.6 | 87.2 | | | 3.4 | 5.9 | 90.7 |
| 19 | 5-leaning left | | | 5.1 | 9.6 | 85.3 | | | 3.7 | 5.2 | 91.1 |
| 20 | 5-leaning right | | | 7.5 | 8.2 | 84.2 | | | 3.8 | 5.8 | 90.4 |

5.2.2. Presence Detection Results of Dynamic Test Cases

Among these dynamic use cases, there are two major categories of tests that are commonly encountered in real life. One category involves the process of getting in and out of the car, in which the tester enters the car at the 10th second and exits at the 50th second. The other category involves large movements in the seat, where the test taker performs upper body swaying and random arm waving for a duration of one minute. Five testers, as listed in Table 2, are used. Each test case lasts for 1 min and is repeated three times. The accuracy is calculated by dividing the number of frames with the correct number of decisions by the total number of decisions, and the final detection results are presented in Table 5.

**Table 5.** Detection results in the dynamic use cases.

| Use Cases | | Detection Accuracy(%) | | | | |
|---|---|---|---|---|---|---|
| Test ID | Number of People Movement | 1 | 2 | 3 | 4 | 5 |
| 1 | 1-Boarding and alighting | 100.0 | | | | |
| 2 | 1-Rocking in the seat | 100.0 | | | | |
| 3 | 2-Boarding and alighting | 3.6 | 96.4 | | | |
| 4 | 2-Rocking in the seat | 1.0 | 97.8 | 1.2 | | |
| 5 | 3-Boarding and alighting | | 3.7 | 93.4 | 2.9 | |
| 6 | 3-Rocking in the seat | | 5.4 | 93.5 | 1.2 | |
| 7 | 4-Boarding and alighting | | | 4.7 | 91.6 | 3.7 |
| 8 | 4-Rocking in the seat | | | 5.1 | 91.7 | 3.2 |
| 9 | 5-Boarding and alighting | | | 2.9 | 8.8 | 88.3 |
| 10 | 5-Rocking in the seat | | | 3.6 | 7.3 | 89.1 |

The results in Table 5 show that there were no incorrect detections of 0 passengers, i.e., the system was able to accurately detect when there were no passengers. The recognition accuracy remained high even during large movements, demonstrating that the state machine-based determination algorithm was able to adapt well to large passenger movements. However, as the number of passengers increased, the detection accuracy decreased, and the accuracy rate for the cases with large movements in the seat was slightly higher than that for getting in and out of the vehicle. This was mainly due to the larger magnitude of movement when getting in and out of the car, which had a slightly greater impact on other positions. The accuracy of the multi-person test cases was higher when compared to the static use case, likely due to the reduced number of weak targets, which is more favorable for radar detection when passengers are not in a completely stationary state. Additionally, the system had an average response time of 1.83 s during passenger changes when getting on and off the vehicle, which met the sensitivity requirement of the detection system.

**6. Conclusions**

This paper focuses on the detection of in-vehicle passengers using FMCW-based radar. The proposed presence detection system addresses the instability caused by passengers' sitting posture and large movements in the vehicle by using a combination of point cloud clustering, post-clustering processing, and a state machine determination algorithm. Experimental results show that the proposed algorithm has a detection accuracy greater than 90.4% in the case of different sitting postures of passengers and a detection accuracy greater than 88.3% in the case of different movements of passengers. At the same time, higher accuracy is obtained in the test cases of multiple people. These results indicate that the proposed methodology is robust and reliable for real-world scenarios of passengers inside a vehicle. However, there is still room for improvement in the detection accuracy for relatively large numbers of passengers. Future research may focus on addressing these issues. This technology can also be extended to other application fields and environments. The proposed system can be combined with robotics and automation systems to assist

people with various disorders, i.e., for detecting the motion of head and limbs, to enhance the mobility of blind people, among other challenging applications where the body motion needs to be detected.

The main challenge for presence detection is the need to improve detection accuracy in scenarios where there are relatively large numbers of passengers in the vehicle (e.g., 4–5 people) or multiple passengers with movements inside the vehicle. Furthermore, the method does not ensure the same level of recognition accuracy when the passenger's back is facing the chip. The primary limitation of this system is that its recognition accuracy is not high enough to be able to detect a crowded condition, i.e., when there are many people in the car, such as with five passengers simultaneously. Additionally, the system has not been tested with more than five people. Future research in this direction is needed.

**Author Contributions:** Conceptualization, Y.L.; Methodology, Y.L. and Y.Q.; Project administration, A.Q.; Software, Y.C. and Y.L.; Supervision, Y.Q.; Validation, Y.C. and Y.L.; Writing—original draft, Y.C.; Writing—review and editing, Y.C., Y.L., J.M., R.H., and F.D.P. All authors have read and agreed to the published version of the manuscript.

**Funding:** This research received no external funding.

**Institutional Review Board Statement:** Not applicable.

**Informed Consent Statement:** Informed consent was obtained from all subjects involved in the study.

**Data Availability Statement:** The data are not publicly available due to privacy reasons.

**Conflicts of Interest:** The authors declare no conflict of interest.

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
