# Peer review of "Non-Contact In-Vehicle Occupant Monitoring System Based on Point Clouds from FMCW Radar"

_technologies, doi:10.3390/technologies11020039_

Round 1

Reviewer 1 Report

Review of paper 2262736

Researching “Non-Contact In-Vehicle Occupant Monitoring System Based on Point Clouds from FMCW Radar” is an indispensable topic in the automobile safety technology because it gives to all those interested to reduce the probability of safety incidents an idea about the aspects which must be considered regarding occupant monitoring using frequency-modulated continuous wave (FMCW) radar. Evolution of vehicle safety and vehicle design are interconnected, both raising challenges due to the interference from passengers' posture, movement, and the presence of multiple people in the cabin.

Introduction

Does the introduction provide sufficient background information for readers not in the immediate field to understand the problem / hypotheses?

Yes. The first section of the paper presents the impact of road traffic accidents and the categories of devices that can detect passengers' presence. Also it presents the methods used for in-vehicle monitoring.

Are the reasons for performing the study clearly defined?

The reasons for developing the research and its objectives for the new design of the system for monitoring vehicle occupants are important for improving the reliability and stability of it.

Are the study objectives clearly defined?

Yes. The main objective is to demonstrate the good performance of the proposed algorithms for occupant detection.

What is the main question addressed by the research?

Can be made an accurate determination of occupants in the vehicle’s cabin?

2. Literature Review and Model Development

Is the literature cited balanced or are there important studies not cited, or other studies disproportionately cited?

The cited literature in this work includes 27 titles and is related to the topics of road safety, Monitoring Indoor People Presence, Gesture Recognition, Radar Applications and 3D Point Cloud Generation.

Please identify statements that are missing any citations, or that have an insufficient number of citations, given the strength of the claim made.

-

Do you consider the topic original or relevant in the field? Does it address a specific gap in the field?

Yes. It is quite authentic. The research makes remarkable progress with the proposed model.

3. Methodology and Data

Are the methodology and data used appropriate to the purpose of the research?

Yes, the methodology is based on “Clustering”, to aggregate point clouds from the same target.

Is sufficient information provided for a capable researcher to reproduce the experiments described?

May be further supplemented.

Are any additional experiments required to validate the results of those that were performed?

Yes. Further experimentation would provide more data regarding the postural kinematics of the passengers.

Are there any additional experiments that would greatly improve the quality of this paper?

Yes. As the technology will gain results it may be improved. Thus experimentation can help.

Are appropriate references cited where previously established methods are used?

Yes

4. Results

Are the results clearly explained and presented in an appropriate format?

Yes

Do the figures and tables show essential data or are there any that could easily be summarized in the text?

Yes.

Are any of the data duplicated in the graphics and/or text?

No.

Are the figures and tables easy to interpret?

May be improved.

Are there any additional graphics that would add clarity to the text?

Surely. The graphs from the experimental part would improve the research definition. The cabin “images” translated into cloud points are showing the steps toward digitization.

Have appropriate statistical methods been used to determine the significance of the results?

Yes.

What does it add to the subject area compared with other published material?

The research makes the correlation between several point clouds for a use case with three passengers in the same row. It shows how Point Cloud Clustering Principle works in practice.

5. Conclusions and Implications

Are all possible interpretations of the data considered or are there alternative hypotheses that are consistent with the available data?

The experiments and their interpretations supported by the results are offered and statistical appreciation is made in the final part.

Are the findings properly described in the context of the published literature?

Yes, in the section dedicated to the Experiment and Evaluation, there are some indications that experiments were carried out using TI's IWR6843AOP [23] for data acquisition but further comparisons are encouraged to be made.

Are the limitations of the study discussed? If not, what are the major limitations that should be discussed?

The limitations of the work may be further described.

What specific improvements should the authors consider regarding the methodology? What further controls should be considered?

The authors could use VR technology for dealing the data.

Are the conclusions consistent with the evidence and arguments presented and do they address the main question posed?

Yes. The conclusion are consistent with the research.

Are the conclusions of the study supported by appropriate evidence or are the claims exaggerated?

Conclusions are supported by the data, demonstrating the importance and opportunity of developing the e detection of in-vehicle passengers using FMCW-based radar. The improvements show that the proposed algorithm has a detection accuracy greater than 90.4% in the case of different sitting postures of passengers and of 88.3% in the case of different movements of passengers.

Are the references appropriate?

Yes

Significance and Novelty

Are the claims in the paper sufficiently novel to warrant publication?

Yes.

Does the study represent a conceptual advance over previously published work?

Yes, it does, by presenting the system for monitoring vehicle occupants with Frequency Modulated Continuous Wave.

Journal Selection

Is the target journal (if known) appropriate? If not, why not?

Yes

What is the likely target audience of this paper?

This scientific paper is useful mainly to safety designers who need specific data regarding passenger positions; to automotive experts and technicians, and active safety experts.

Minor comments

Please refer to the comments in the edited manuscript file for minor comments.

Accept if minor revisions are made.

Major comments

To publish this paper in your target journal, the following revisions are strongly advised:

The English language and style are fine/minor spell check required.

Additional comments on the tables and figures.

There are no additional comments.

Reviewer 2 Report

The paper is devoted to the problem of detecting and determining the number of passengers inside a car using FMCW Radar. The authors develop existing methods for determining the number of passengers based on clustering methods. Clustering methods are applied to reflected pulses received by Radar. The authors propose two approaches to improve the accuracy of determining the number of passengers. The first is post-processing of clustering results, as a result of which some clusters are excluded from consideration, other clusters are combined, and some clusters are divided. The second approach is to fix the moments of embarkation and disembarkation of passengers and is based on a state machine.

The paper provides a detailed description of the use of FMCW Radar to determine the number of passengers in a car. The methods proposed by the authors are described, and the results of testing these methods are presented. The article also provides an overview of other approaches to determining the number of passengers.

The article is clear, relevant for the field and presented in a well-structured manner. The literature cited is up-to-date and relevant. The authors do not include excessive self-citations. The article sounds scientific. The results obtained are reproducible. Figures and diagrams are appropriate. The conclusions are consistent with the evidence and arguments presented.

The article can be published after minor revision.

1) Section 4.2 should indicate how the x,y and z axes are oriented with respect to the vehicle.

2) In section 5.1, authors should indicate which tasks were solved by MATLAB and Python library functions, and which were solved using the original code.

Reviewer 3 Report

The authors address the issue of in-vehicle occupant detection by exploiting FMCW radar. The introducing section and all the contribution is quite clear. However the author should better explain what is the novelty of the paper compared to the existing literature.

Also Section 5 can be improved. Indeed the authors do not show radar measurement but only the overall results. It is a paper based on radars, it can be interesting evaluating the measured data by showing some examples.

The authors write: “The main contribution of this paper is to filter, segment and merge the point cloud cluster according to the actual distribution of car seats, aiming at the impact of passenger sitting movements and multi-person environment on the point cloud clustering, so as to make the point cloud can correctly correspond to the actual target, and thus improve the recognition accuracy in the real in-vehicle environment.”. It seems that the main contribution of this paper consists only in re-adapting existing procedure to the in-cabin scenario that of course it is not enough to publish on this journal.

In Section 2, I suggest to mention also this very recent work focused on this topic:

E. Cardillo, L. Ferro and C. Li, "Microwave and Millimeter-Wave Radar Circuits for the Next Generation Contact-Less In-Cabin Detection," 2022 Asia-Pacific Microwave Conference (APMC), Yokohama, Japan, 2022, pp. 231-233, doi: 10.23919/APMC55665.2022.9999764.

Moreover, you can also consider that similar techniques are also applied in the field of head motion detection for assisting people with different disorders or also to enhance the mobility of blind people or in HVAC system (due to the need to detect the presence of occupants).

It seems that your system does not consider techniques like vital sign or target micro-Doppler signature detection which can enhance the detection. It is not clear if your system can be considered robust compared to work exploiting these techniques (again you can find a lot of very recent example).

Round 2

Reviewer 3 Report

The authors addressed all my concerns.